

# Increased vitamin D intake may reduce psychological anxiety and the incidence of menstrual irregularities in female athletes

Mana Miyamoto[1,2,3], Yuko Hanatani[1] and Kenichi Shibuya[1,2,3]

[1] Japan Rowing Association, Tokyo, Japan
[2] Graduate School of Health and Welfare, Niigata University of Health and Welfare, Niigata, Japan
[3] Department of Health and Nutrition, Niigata University of Health and Welfare, Niigata, Japan

## ABSTRACT

**Background:** Vitamin D deficiency has been associated with major depression and premenstrual mood symptoms, and menstrual irregularity has been correlated with mental anxiety. However, the potential effect of increased vitamin D intake on reducing the risk of menstrual irregularities by decreasing psychological anxiety is yet to be fully elucidated. The existence of such a relationship in athletes with high levels of psychological anxiety and adequate dietary intake remains unknown. Therefore, this study aimed to examine the effects of vitamin D intake on psychological anxiety levels and the risk of menstrual irregularities in healthy college-and international-level female athletes.

**Methods:** Female intercollege-level track and field and international-level rowing athletes ($n$ = 107) aged 15–24 years were included in this study. Their nutritional intake, body mass, body fat, mental anxiety, and menstrual irregularities were investigated. A generalized linear mixed model (GLMM) was used to examine the effects of several parameters on menstrual irregularities. The independent variables introduced into the GLMM were determined based on Akaike's information criterion.

**Results:** The GLMM identified a significant interaction effect of vitamin D intake and state anxiety on menstrual irregularities, with a $p$-value of 0.049 and an odds ratio of 0.423. The study results suggest that increased vitamin D intake in relatively young endurance athletes may reduce mental anxiety, consequently decreasing menstrual irregularities.

## INTRODUCTION

Recent reports show that many athletes experience menstrual disorders, which may have serious consequences in sports (*Gulliver et al., 2015*; *Miyamoto, Hanatani & Shibuya, 2021a*). Approximately 20% of elite female rowers experience a mental disorder, and mental disorders, such as anxiety, are significantly related to menstrual irregularities (*Miyamoto, Hanatani & Shibuya, 2021a*; *Thomas, Erdman & Burke, 2016*). However, menstrual irregularities can also be caused by energy deficiency (*Miyamoto, Hanatani & Shibuya, 2021b*). Chronic energy deficiency is associated with the impairment of various physical functions, such as estradiol and progesterone suppression (*Loucks, 2013*).

Corresponding author
Kenichi Shibuya,
shibuya@nuhw.ac.jp

In contrast, our previous study suggested that increased mental anxiety may play a greater role in the development of menstrual irregularities in elite female athletes than nutritional status (*Miyamoto, Hanatani & Shibuya, 2021a*). Athletes with greater mental anxiety, particularly state anxiety, have an increased likelihood of experiencing menstrual irregularities (*Miyamoto, Hanatani & Shibuya, 2021a*). Approximately 46% of elite athletes have considerable mental health problems that require clinical diagnosis (*Eskandari et al., 2007*). Athletes experience increased mental anxiety and stress compared to the general population, which may have a greater impact on menstrual function. Therefore, preventing this increased mental anxiety and stress is important for maintaining a healthy physical condition, particularly a normal menstrual function. One possible method to reduce mental anxiety and stress is to increase vitamin D intake. Vitamin D receptors are present throughout the brain, and its deficiency has been associated with adverse effects on the central nervous system (*McCann & Ames, 2008*). Vitamin D receptors and activating enzymes are prominent in the hypothalamus and substantia nigra, and are involved in glucocorticoid signaling in hippocampal cells. Moreover, its deficiency has been correlated with major depression (*Eskandari et al., 2007*) and premenstrual mood symptoms (*Thys-Jacobs et al., 1995*) in women and mood and cognitive disorders in older adults (*Wilkins et al., 2006*). Two randomized controlled trials have demonstrated that increasing vitamin D levels improve symptoms of depression (*Gloth, Alam & Hollis, 1999*; *Jorde et al., 2008*). However, the relationship between vitamin D intake and mental health/menstrual function in physically healthy athletes experiencing significant mental anxiety and stress remains unclear. The hypothesis of the present study was that vitamin D intake itself might not affect menstrual irregularities, but that vitamin D intake might reduce psychological anxiety, which in turn might reduce the probability of menstrual irregularities. Therefore, this study aimed to examine the impact of vitamin D intake on psychological anxiety levels and menstrual function in healthy college- and international-level female athletes in cross-sectional investigation.

## MATERIALS AND METHODS

### Participants

The study population comprised 107 female athletes aged 15–24 years who had no neurological and metabolic disorders, 41 inter-college-level track and field athletes, and 66 international-level rowing athletes. Rowing athletes were recruited from the national team members of the Japan Rowing Team, and Track & Fields athletes were recruited from the member of the Track & Field team of Niigata University of Health and Welfare. All athletes had a body mass index (BMI) of >18.5. Data from each athlete were obtained from April 2021 to February 2022.

The appropriate sample size ($n = 107$) was calculated using the following: odds ratio = 2, H0 = 0.15, $\alpha = 0.05$, power = 0.8, and X distribution = normal on logistic regression. The Ethics Committee of Niigata University of Health and Welfare approved this study (approval #17982-180606). After receiving a detailed explanation of the nature of the study and its noninvasiveness, each participating athlete provided written informed consent.

**Table 1 Mean values of physical characteristics and state anxiety (Mean ± S.E).**

| Height (cm) | Body mass (kg) | %BF (%) | BMI |
|---|---|---|---|
| 165.6 ± 0.6 | 59.0 ± 0.6 | 19.3 ± 0.4 | 21.5 ± 0.2 |

| State anxiety | Trait anxiety | | |
|---|---|---|---|
| 42.0 ± 0.9 | 45.4 ± 0.9 | | |

Note:
  BF, body fat; BMI, body mass index.

## Dietary intake, body mass, and body composition

The participants tracked their dietary intake using a meal recording method (cf.: *Miyamoto, Hanatani & Shibuya, 2021a*, *2021b*). Athletes sent a picture of each meal (breakfast, lunch, dinner, and snacks) for 3 days (one non-training day and two training days) each week to each team dietitian. The dietitians analyzed each nutrient intake from these pictures. Nutrient information was obtained from the Japan National Nutrient Database or product-specific nutrition fact panels. Average daily nutrient intake (total energy intake (kcal), macronutrients (g), and micronutrients (mg and μg)) was determined. Body mass (BM) and body fat percentage (%BF) were measured using a commercially available home scale (BC-314; Tanita. Co., Tokyo, Japan) as validated in a previous study (*Hicks et al., 2017*). The BMI was calculated as weight (kg)/height$^2$ (m$^2$). Height, BM, and %BF were measured once during the investigation.

## State and trait anxiety and menstrual cycle

State and trait anxiety were measured using the State and Trait Anxiety Inventory (STAI) (*Spielberger, Gorsuch & Lushene, 1970*). Participants tracked and reported their menstrual cycles (using a paper/pen calendar). State and trait anxiety were measured once during the investigation; *Spielberger, Gorsuch & Lushene (1970)* defined state anxiety as a temporal reaction to an adverse event and trait anxiety as the tendency to respond with concerns and worries to various situations.

## Statistical analyses

Data are expressed as mean ± standard error. Statistical analyses were conducted using the lmerTest package in the R software (*Kuznetsova, Brockhoff & Christensen, 2017*). The Akaike's information criterion (AIC) was used to validate the generalized linear mixed model (GLMM) parameters. Based on the AIC score, a GLMM was performed using age, body weight, and trait anxiety as random effects and vitamin D intake, state anxiety, and the interaction between vitamin D intake and state anxiety as fixed effects. The significance level was set as 0.05.

# RESULTS

## Physical characteristics

The mean %BF was 19.3%, with a 95% confidence interval (CI) of [18.5–20.0] (Table 1). Of the 107 athletes, 50 had a %BF of <20%. The mean BMI was 21.5, with a 95% CI of [21.1–21.8]. None of the participants had a BMI of <18.5. Therefore, the energy intake of
**Table 2 Mean values of energy and macronutrient intake (Mean ± S.E.).**

| Energy (kcal) | Protein (g) | Fat (g) | CHO (g) |
|---|---|---|---|
| 2487.6 ± 52.5 | 108.5 ± 2.5 | 81.3 ± 1.9 | 321.1 ± 8.8 |

Note:
CHO, Carbohydrate.

**Table 3 Mean values of micronutrient intake (Mean ± S.E.).**

| Ca (mg) | Fe (mg) | VA (µg RAE) | VB$_1$ (mg) |
|---|---|---|---|
| 786.2 ± 32.9 | 12.4 ± 0.4 | 1203.1 ± 158.0 | 1.7 ± 0.1 |
| **VB$_2$ (mg)** | **VC (mg)** | **VD (µg)** | |
| 2.1 ± 0.1 | 198.8 ± 13.8 | 8.8 ± 0.6 | |

Note:
Ca, calcium; Fe, iron; VA, vitamin A; VB$_1$, vitamin B$_1$; VB$_2$, vitamin B$_2$; VC, vitamin C; VD, vitamin D.

the participants was considered relatively adequate. The mean state anxiety value was 42.0, with a 95% CI of [40.2–43.8]. Sixty-seven athletes with a clinical cutoff value of >40 points (*Grant, McMahon & Austin, 2008*) exhibited state anxiety. The mean state anxiety value was 45.4, with a 95% CI of [43.5–47.2]. Seventy-five athletes with a clinical cutoff value of >40 points (*Grant, McMahon & Austin, 2008*) exhibited trait anxiety.

## Macronutrient intake

The mean energy intake was 2,487.6 kcal (95% CI [2,384.7–2,590.5] kcal). The mean protein intake was 108.5 g (95% CI [103.6–113.3] g), and the mean protein intake per body mass (Pro/BM) was 1.8 g/kg (95% CI [1.8–1.9] g/kg) (Table 2). Pro/BM values ≥1.0, >1.5, and >2.0 g/kg were observed in 107, 81, and 40 athletes, respectively. The mean carbohydrate (CHO) intake was 321.1 g (95% CI [303.9–338.3] g), whereas the mean CHO intake per body mass (CHO/BM) was 5.5 g/kg (95% CI [5.2–5.7] g/kg). CHO/BM values >5 and 6 g/kg were observed in 57 and 33 athletes, respectively.

## Micronutrient intake

The mean iron intake was 12.4 mg (95% CI [11.6–13.1] mg). Iron intake of <10, 15, and 18 mg was observed in 34, 82, and 96 athletes, respectively (Table 3). The mean calcium intake was 786.2 mg (95% CI [721.8–850.7] mg). Calcium intake of >900 and 1,200 mg was observed in 30 and 14 athletes, respectively. The mean vitamin D intake was 8.8 µg (95% CI [7.7–10.0] µg).

## GLMM analysis

The independent variables introduced into the GLMM were determined based on the AIC. GLMM identified significant factors related to menstrual irregularity, including vitamin D intake and state anxiety ($p = 0.913$ and $p = 0.009$, respectively) (Table 4). The odds ratios for vitamin D intake and state anxiety were 0.968 and 2.687, respectively.

**Table 4 Variables and estimates of each variable of the multiple logistic models.**

| Variables | Estimate ± S.E. | z value | p value | O.R. | 95% CI for O.R. |
|---|---|---|---|---|---|
| Vitamin D | −0.032 ± 0.297 | −0.110 | 0.913 | 0.968 | [0.540–1.734] |
| State anxiety | 0.988 ± 0.378 | 2.614 | 0.009 | 2.687 | [1.280–5.639] |
| Vitamin D * State anxiety | −0.861 ± 0.437 | −1.968 | 0.049 | 0.423 | [0.180–0.996] |

**Note:**
O.R., odds ratio.

## DISCUSSION

The study results revealed that vitamin D may effectively reduce mental anxiety and that its high intake may decrease the incidence of menstrual irregularities by reducing mental anxiety. High state anxiety is associated with an increased incidence of menstrual irregularities (*Miyamoto, Hanatani & Shibuya, 2021a*), and vitamin D intake is associated with an increased incidence of depression (*Eskandari et al., 2007*), premenstrual syndrome (*Thys-Jacobs et al., 1995*), and cognitive impairment (*Wilkins et al., 2006*). To the best of our knowledge, this study is the first to suggest that high vitamin D intake may reduce mental anxiety and, consequently, decrease the risk of menstrual irregularities.

### Energy intake

The average energy intake, weight, and %BF of the participants were 2,487.6 kcal/day, 59.0 kg, and 19.3%, respectively. Based on these results, the energy required to maintain healthy physical function was 2,142 kcal, calculated from the energy availability (EA) level (>45 kcal/kg fat-free mass/day) (*Thomas, Erdman & Burke, 2016*). Considering the energy requirements of the participants, owing to their exercise regime, the energy intake in this study was similar to the required energy intake. In addition, the results of the model selection by AIC showed no relationship between menstrual irregularities and energy intake. The mean BMI of the participants was 21.5, and none had a BMI of <18.5; 26 athletes had a BMI of <20. The significantly inadequate nutritional intake of a few participants may have influenced the study results.

### Protein and CHO intake

The mean protein intake of participants was 108.5 g/day. Eighty-one and 40 of the 107 participants consumed 1.5 and 2.0 g of Pro/BM, respectively. Thus, the athletes consumed an adequate amount of protein (*Thomas, Erdman & Burke, 2016*). The mean CHO intake of participants was 321.1 g/day. Of the 107 participants, 57 and 33 consumed 5 and 6 g of CHO/BM. However, only 19 participants consumed >7 g of CHO/BM. Menstrual irregularities increase in elite female athletes with inadequate CHO intake (*Miyamoto, Hanatani & Shibuya, 2021b*). However, the results of the model selection by AIC in this study showed no relationship between menstrual irregularities and CHO intake. Therefore, further research is required to determine the level at which CHO intake should be maintained to reduce the incidence of menstrual irregularities.

### State and trait anxiety inventory and vitamin D

In this study, the means for state and trait anxiety were 42.0 and 45.4, respectively, above the clinical cutoff of 40 points (*Grant, McMahon & Austin, 2008*). Of the 107 athletes, 67 had state anxiety, and 75 had trait anxiety above 40 points. These values were much higher than those observed in the general population (19.5–20.1%) (*Zsido et al., 2020*), indicating that athletes are mentally unstable. It has also been reported that high mental anxiety can cause menstrual irregularities (*Miyamoto, Hanatani & Shibuya, 2021a*). Previous studies have demonstrated that vitamin D intake is effective against anxiety and stress caused by mental instability (*Wilkins et al., 2006*; *Eskandari et al., 2007*). Similarly, this study suggests that a relationship may exist between low vitamin D intake and high mental anxiety and that high anxiety may cause menstrual irregularities. Therefore, the results of this study corroborate the findings of previous studies (*Eskandari et al., 2007*; *Wilkins et al., 2006*) on a relationship between vitamin D intake and mental anxiety/stress levels. To the best of our knowledge, this is the first report to demonstrate the relationship between vitamin D levels, mental anxiety, and menstrual irregularities in athletes, including high-level athletes. The mean vitamin D intake in this study was 8.8 μg/day. As shown in Table 4, the results suggest that high vitamin D intake may reduce state anxiety and decrease menstrual irregularities. The STAI (used as a clinical indicator (*Grant, McMahon & Austin, 2008*)) was used to determine state/trait anxiety scores in this study. The participants included athletes who scored above the clinical cut-off of 40 points (derived from a wide range of STAI data). Given that the GLMM estimates for STAI data were calculated using a wide range of STAI data, the study results are considered valid to a certain degree. The cross-sectional study offers generic insights and could be improved with a study with an experimental sample to which predetermined doses of vitamin D should be administered through the diet. Other measures of mental stress should be used to further clarify the relationship between state anxiety and menstrual cycle irregularity. In addition, quantifying the hormones involved in menstrual irregularities and clarifying the physiological effects of mental stress and anxiety on menstrual irregularities can provide a more accurate understanding of the menstrual cycle. Similarly, future research should focus on measuring blood vitamin D levels and obtaining more accurate data on vitamin D intake.

## CONCLUSIONS

This study aimed to verify the hypothesis that increased vitamin D intake is associated with reduced mental anxiety, which may decrease the occurrence of menstrual irregularities in young female athletes. The study results suggest that increased vitamin D intake in relatively young endurance athletes aged 15–24 years may reduce mental anxiety, consequently lowering the incidence of menstrual irregularities. These findings may contribute to future developments in this area of research. However, future longitudinal studies will provide a more accurate interpretation of the data from this study. Generating large-scale data would also promote a more accurate understanding of the impact of mental anxiety on menstrual cycle irregularities. Finally, the psychological and molecular basis of menstrual irregularities should be studied extensively.

### Funding

There was no funding for this research.

### Competing Interests

Kenichi Shibuya is an Academic Editor for PeerJ.
Yuko Hanatani is an employee of Japan Rowing Association.

### Author Contributions

- Mana Miyamoto conceived and designed the experiments, performed the experiments, analyzed the data, prepared figures and/or tables, authored or reviewed drafts of the article, and approved the final draft.
- Yuko Hanatani conceived and designed the experiments, performed the experiments, authored or reviewed drafts of the article, and approved the final draft.
- Kenichi Shibuya conceived and designed the experiments, analyzed the data, prepared figures and/or tables, authored or reviewed drafts of the article, and approved the final draft.

### Human Ethics

The following information was supplied relating to ethical approvals (*i.e.*, approving body and any reference numbers):

The Ethics Committee of Niigata University of Health and Welfare provided approval to perform this study (Approval #17982-180606). After receiving a detailed explanation regarding the nature of the study and its noninvasiveness, each participating athlete provided written informed consent.

### Data Availability

The raw data is available in the Supplemental Files.

### Supplemental Information

Supplemental information for this article can be found online at http://dx.doi.org/10.7717/peerj.14456#supplemental-information.

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
