# Peer review of "Increased vitamin D intake may reduce psychological anxiety and the incidence of menstrual irregularities in female athletes"

_PeerJ, doi:10.7717/peerj.14456_

## Round 0.1 · original submission · Major Revisions

Dear authors,

Please reply point by point to the reviewer's comments.
Follow the reviewers' suggestions carefully.
The manuscript needs major revision.

·

Basic reporting

The present study aimed to examine the impact of vitamin D intake on psychological anxiety
levels and menstrual function in healthy college- and international-level female athletes.
The topic is of considerable interest and is treated with a sufficiently updated background with citations of supporting studies to the aims of the study.
The authors describe numerous limitations of the study and also do so in the description of the sample as well as in the dedicated paragraph. for example they admit that "The fact that few athletes in the population in the present study had significantly inadequate nutritional intake may have influenced these results this study".
It would therefore seem that the factor in energy intake has no influence on menstrual disorders but this is only evident due to a bad criterion for including the sample.
If you want to improve the quality of the study about this important objective (energy intake) it would be advisable to exclude BMIs less than 18.5 and recalculate the statistics even if on a small sample, however, possible future longitudinal studies are highlighted that could better express the correlation between vitamin D anxiety and menstrual disorders.
The results of the present study therefore focused on increasing vitamin D intake relatively young endurance athletes between the ages of 15 and 24. This increase can reduce mental anxiety and thus lead to a lowering incidence of menstrual irregularities. Even the study of the State Anxiety alone is limiting for a complete analysis of the neurohormonal factors most related to stress, this limitation is also considered by the authors. The cross-sectional study offers generic insights and could be improved with a study with an experimental sample to which predetermined doses of vitamin D should be administered through the diet. However, the correlation with anxiety measurement is significant and encourages better controlled future studies.

Experimental design

The cross-sectional study offers generic insights and could be improved with a study with an experimental sample to which predetermined doses of vitamin D should be administered through the diet. However, the correlation with anxiety measurement is significant and encourages better controlled future studies. It would therefore seem that the factor in energy intake has no influence on menstrual disorders but this is only evident due to a bad criterion for including the sample.
If you want to improve the quality of the study about this important objective (energy intake) it would be advisable to exclude BMIs less than 18.5 and recalculate the statistics even if on a small sample, however, possible future longitudinal studies are highlighted that could better express the correlation between vitamin D anxiety and menstrual disorders.

Validity of the findings

The results of the present study therefore focused on increasing vitamin D intake relatively young endurance athletes between the ages of 15 and 24. This increase can reduce mental anxiety and thus lead to a lowering incidence of menstrual irregularities. Even the study of the State Anxiety alone is limiting for a complete analysis of the neurohormonal factors most related to stress, this limitation is also considered by the authors. The authors describe numerous limitations of the study and also do so in the description of the sample as well as in the dedicated paragraph. for example they admit that "The fact that few athletes in the population in the present study had significantly inadequate nutritional intake may have influenced these results this study".

Additional comments

Lines 54 and 57 would unify them and eliminate the points you see in the template.

Reviewer 2 ·

Basic reporting

- The introduction is confusing and not smoothly to read. I suggest reviewing this section entirely and linking the sentences together according to the rationale of the study.
- The introduction does not include all the references necessary to cover the background provided by the authors. For example: (Line 48-49): “Recently, however, many athletes experience menstrual disorders, which may have serious consequences in the field of sports”; (Line 51-52): “Menstrual irregularity can be caused by energy deficiency, hormonal changes, and increased mental anxiety”; (Line 62-63): “Athletes with greater mental anxiety, particularly state anxiety, have an increased likelihood of experiencing menstrual irregularities” and so on.
- Authors should clearly define their hypothesis.

Experimental design

- Please define the study design.
- Authors should specify how sample size was determined (tail(s), effect size, α err prob).
- The study does not provide sufficient details to allow for its replicability. Below some comments:
- Please report how the participants were recruited and the sampling method.
- Eligibility criteria for participants should be clearly reported.
- The setting of data collection should be reported in detail.

Validity of the findings

- Data interpretation should be consistent with results found, considering other relevant evidence not only in agreement but also in contrast with them.

---

## Round 0.2 · accepted · Accept

Dear authors,

Congratulations! The manuscript has greatly improved and is now worthy of publication.

Good luck with your future projects!

Reviewer 2 ·

Basic reporting

No comment

Experimental design

No comment

Validity of the findings

No comment

Additional comments

No comment